Corrected: Publisher correction

# A multistage rotational speed changing molecular rotor regulated by pH and metal cations

Yingying Wu[1], Guangxia Wang[1,2], Qiaolian Li[1,2], Junfeng Xiang[2], Hua Jiang[1] & Ying Wang [1]

Despite having significant applications in building nanomachines, molecular rotors with the rotational speed modulations to multiple stages in a wide range of frequency have not yet been well established. Here, we report the discovery of a stimuli-responsive molecular rotor, the rotational speed of which in the slow-to-fast range could be modulated to at least four stages triggered by acid/base and metal cations. The rotor itself rotates rapidly at ambient or elevated temperature but displays a restricted rotation after deprotonation due to the produced intramolecular electrostatic repulsion. Subsequent addition of $Li^+$ or $Na^+$ cations introduces an electrostatic bridge to stabilize the transition state of the deprotonated rotor, thus giving a cation-radius-dependent acceleration of the rotation to render the rotor running at a mid-speed. All the stimuli are highly reversible. Our studies provide a conceptual approach for constructing multistage rotational-speed-changing molecular rotors, and further, the practical nanomachines.

---

[1] College of Chemistry, Beijing Normal University, 100875 Beijing, China. [2] Institute of Chemistry, Chinese Academy of Sciences, 100190 Beijing, China. Correspondence and requests for materials should be addressed to Y.W. (email: ywang1@bnu.edu.cn)

Nanomachines performing similar functions as macroscopic machines have broad potential applications in sensing[1], medicines[2], and smart materials[3], but the current top-down fabrication approach towards them faces huge obstacles due to the intrinsic limitation of miniaturization technology[4]. Inspired by the remarkably precise manipulations in movements of biological motors[5–7], scientists have started to develop such machines from molecular-level motors[4,8–12]. Such a bottom-up approach provides the possibility of constructing complicated nanomachines, but, at the same time, it critically relies on the precision and effectiveness on the motion control of the built molecular motors. In this regard, exciting achievements have been recently made in controlling the motional direction[13,14] and modes[15,16] of molecular motors. Nevertheless, the control over motional speed is still limited and unsatisfactory due to the lack of versatility of the motor speed[17–22].

Rotary parts or components are indispensable in all macroscopic machines. As such, speed regulation is typically a basic requirement but has never been a problem using the modern technology. For example, the rotational speed of the rotational systems in cars and electric fans can be finely tuned to several different stages, typically including a stage of a full stop, several stages in the mid-speed range and a high (maximum)-speed stage. The macroscopic rotary parts are extended conceptually to the molecular level as molecular rotors, which possess a rotator component that can rotate of a stator around a common axis[23–28]. Ideally, the molecular rotors are similarly of multistage rotational rates that overall cover a wide range of frequency to enable them to operate at slow ($<10^{-1}$ Hz), intermediate ($10^{-1}$–$10^3$ Hz), fast ($10^3$ – $10^7$ Hz) or ultrafast ($>10^7$ Hz) speeds as required. However, speed regulation for such species are found to be extremely challenging. To date, the control of the dynamics of molecular rotors in most cases were focused on stimuli-responsive molecular brakes, within which intramolecular rotation is hindered or constrained as a response to external chemical[29–31], electrochemical[32–35], or photochemical[36–39] stimuli. Several isolated cases involve accelerating the internal rotation[40–44]. Despite of providing conceptual approaches on the rotation regulation, these ON/OFF-type systems work only at a fixed rotational speed at a given temperature, which failed to match the various operational mode in most of their macroscopic counterparts.

It is only until recently that researchers have started to design rotor systems with the rotational speed of precise regulation. Groundbreaking studies by Sozzani and coworkers in this field identified that rotational speeds of the rotors built in porous materials could be finely tuned by the influence of the uptaken guest molecules on the free volumes of the rotors[18,19]. Horike and coworkers demonstrated that such precise regulation could be achieved using a solid-solution approach as well[20]. Though generally provided an effective way to regulate the rotation, the rotational speed of such rotary systems could only be modulated in the ultrafast-to-fast range (from $10^8$ to $10^4$ Hz in frequency). Recently, Feringa and Wezenberg showed that, by modulating steric hindrances of the rotary system using metal ions as allosteric effectors, rotational speed of a rotor incorporating 4,5-diazafluorenyl coordination motif could be finely tuned into four stages[21]. Nevertheless, for this system, the rotational frequency could just be tuned in the range of $8.2 \times 10^{-3}$–0.25 Hz. More recently, Shimizu verified that apparent rotational barrier of a rotor could be tuned by using guests with different H-bonding abilities to disrupt the intramolecular H-bond that catalyzes the rotation of the rotor, but corroboration may be still required to identify whether addition of the guests changed the rotational speed of the rotor or just the ratio of the rotor molecules at the native state to those with the guest H-bonded[22]. All these indicate

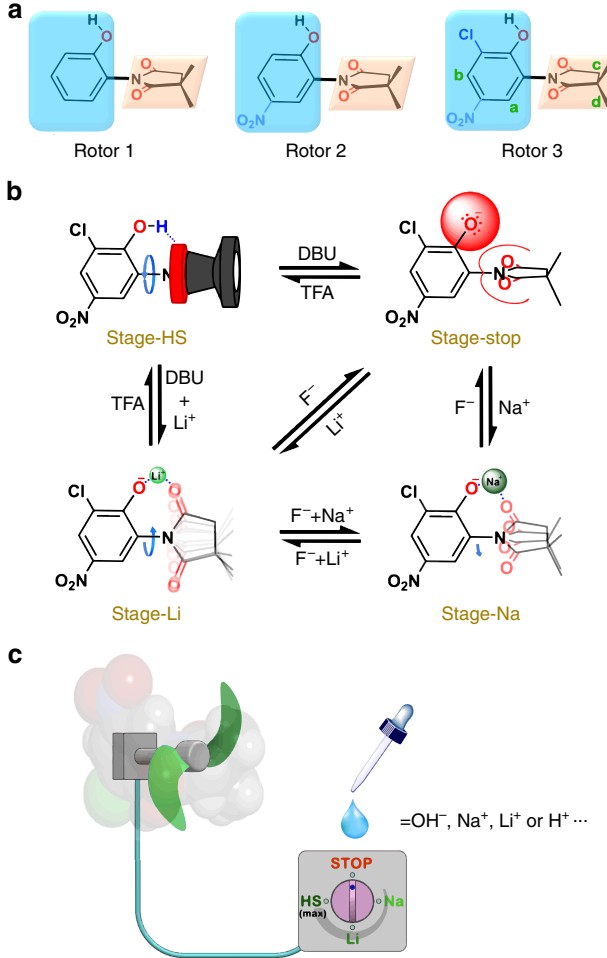

**Fig. 1** Overview of the multistage rotational speed changing molecular rotors. **a** Chemical structures of the three studied molecular rotor candidates **1**–**3**. The difference in structures lies in the substituents on the phenol stators. **b** Schematic representation of the rotor **3** rotating at speed in four different stages and the interconversions between them. The stimuli we utilized are shown nearby the corresponding black arrows. The blue curving arrows represent the rotational speeds (but not the directions) of the rotor under different conditions, being faster when a longer arrow is used. **c** Cartoon illustration of the molecular rotor with the rotational speed tunable within four stages via chemical stimuli. The overall dynamic properties make the rotor a good mimic of macroscopic electric fan

that the precise modulation of the rotation of synthetic molecular rotors in a wide range of frequency, especially fine-tuning of the rotational speed in the range from slow (or stop) to ultrafast ones to several differentiable stages, as those we can often observed in macroscopic rotary machines, still remains a very tough challenge.

In this contribution, we report the discovery of a molecular rotor system that exhibits multistage rotational-speed-changes in the slow-to-fast range of frequency at one given temperature, wherein the rotation is regulated by the addition (or extraction) of acid/base (pH) and metal cations (Fig. 1). The screened rotor consists of a succinimide rotator and a stator of phenol with the O–H group polarized (Fig. 1a), which rotates fast at the native state but revolves slowly upon deprotonation ascribed to the produced electrostatic repulsion between the phenolate anion and imide carbonyls. Subsequent addition of metal cations resumes the rotation with the acceleration amplitude dependent on radius of the cations added, as a result of an induced electrostatic bridge

to lower the energy level of the transition state (Fig. 1b). All the stimuli-responses exhibit a reversible manner, enabling the molecular rotor to compellingly mimic the macroscopic electric fans (Fig. 1b, c).

## Results

**Design and synthesis.** In an effort to design molecular rotors with finely tunable speed, we were drawn to the unique advantages of chemical stimuli. Most chemical stimuli are easily triggered, and some can also be eliminated facilely, which is ideal for constructing stimuli-responsive switches[45,46]. For example, we have reported the acid/base stimuli to control the configuration/conformation[47] or the dynamics[15] of molecules, in addition to metal cations discovered by other researchers with similar purpose[48]. In particular, Kubik's and Feringa's studies have demonstrated that chemical stimuli can be used for rotor speed regulations. On the other hand, at a given temperature, rotational speed of molecular rotors is determined entirely by the rotational barrier, i.e., the energy gap between the transition state conformation and the ground state of the molecules. Consequently, rotational frequencies could be increased upon decreasing the energy barrier either by lowering the energy level of the transition state (TS) or by destabilizing the ground state (GS) of the molecule. As for the former approach, while Kubik's and Feringa's studies hinted that changing the degree of steric crowding in the motion region is not effective enough for fine-tuning rotor speed in a wide range with practical application value, the method of forming intramolecular interactions between the stator and the rotator provides another option[41,43]. Rebek has reported that metal complexation and protonation gave stabilizing the TS of a bridged bipyridyl derivative thus being able to modulate the racemization rate of the bipyridyl[49]. We envisioned that chemical stimuli capable of introducing non-covalent bonding between the two components of rotors, with varying bond strengths, to stabilize the rotational transition state in varying degrees for precisely tuning the rotation of the molecules.

In this proof-of-concept study, we designed three simple molecular rotor candidates **1**–**3** (Fig. 1a), all of which contain a 3,3-dimethylsuccinimide rotator and a phenolic backbone serving as the stator. The 3,3-dimethylsuccinimide is an excellent moiety to construct molecular rotor due to its shape-persistent structure, in which the two diastereotopic methyl groups are excellent probes for detecting the intramolecular rotations as reported by Horike and others[43,44]. To explore the effect of acidity on the dynamics of the molecules, the rotors were designed to possess different substituents on the phenolic stator: rotor **1** carries only a bare phenolic group; rotor **2** has an extra nitro substituent at para-position and **3** possesses an additional chlorine atom at the ortho-position further (Fig. 1a). Consequently, the acidity of the molecules increases in the order of **3** > **2** > **1** (as a reference, the p$K$a of phenol, 4-nitrophenol, and 2-chloro-4-nitrophenol is 9.95, 7.14, and 5.45, respectively)[50].

Molecules **1**–**3** were prepared by the imidation of the corresponding 2-aminophenol or its derivatives with 2,2-dimethylsuccinic anhydride (Supplementary Fig. 1 and Supplementary Methods 1 and 2). The structures of the molecules were unambiguously confirmed by the assignment of their experimental $^1$H and $^{13}$C nuclear magnetic resonance (NMR) spectra in acetonitrile-$d_3$ using standard 2D NMR spectroscopy (Supplementary Figs. 7–18 and 149–154), the correlation between the DFT-calculated and experimental $^1$H and $^{13}$C NMR chemical shifts (Supplementary Table 2–4) as well as the high-resolution mass spectrometry (Supplementary Figs. 2–4).

The single crystals of **1** – **3** were obtained by slowly evaporating a solvent mixture of ethyl acetate and hexane at ambient

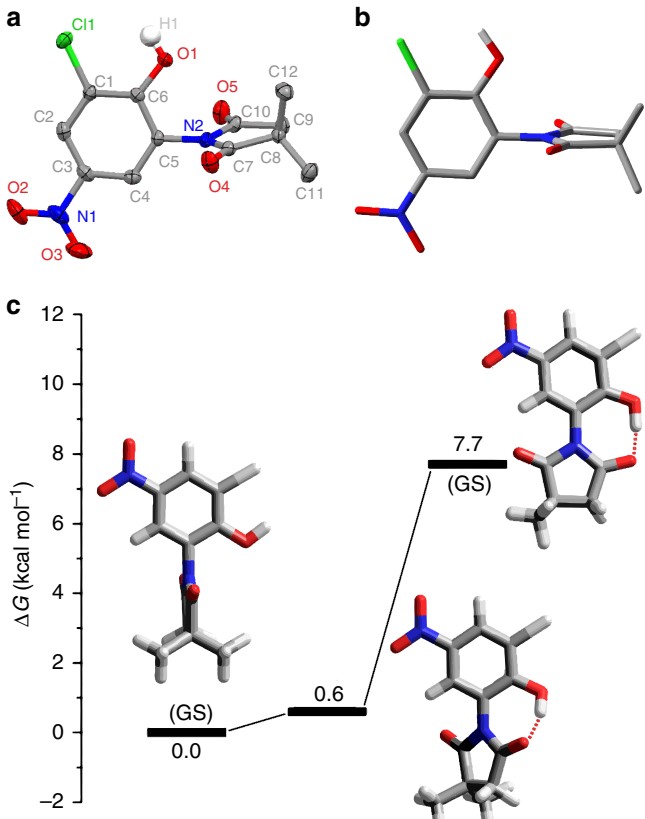

**Fig. 2** Structural geometries of **3** at different states. **a** The X-ray molecular structure of **3**. Atoms are depicted with thermal ellipsoids set at the 50% probability level. All hydrogen atoms except the hydroxyl one are omitted for clarity. **b** Energy-minimized geometry (DFT B3LYP/6-31+G(d,p)/IEF-PCM-UFF) of **3** in acetonitrile. **c** Plots of the relative energies between the ground state and transition state geometries of **3** calculated at DFT B3LYP/6-31+G(d,p)/IEF-PCM-UFF level of theory. Further details are reported in Supplementary Table 24 and 25

temperature (Supplementary Method 3). The X-ray structures of all molecules showed an expected nearly coplanar arrangement of the succinimide rings due to the $sp^2$ hybridization of the imide, located in a plane perpendicular to that of the phenolic stators (Fig. 2a, Supplementary Figs. 5–6 and Supplementary Table 1). A slight distortion of the succinimide rings were also observed (Supplementary Table 26), which may be resulted from packing effects. The hydroxyl group in this state is too far to H-bond with the carbonyls and thus directed exo.

DFT calculations (Supplementary Method 12)[51] carried out at the B3LYP/6-31+G(d,p) level of theory confirmed the perpendicular conformation (Fig. 2b and Supplementary Tables 24 and 29) as the globe minimum for all the molecular rotors in two polar solvents, including acetonitrile and acetone, with very low energy gaps (<1 kcal mol$^{-1}$) between this conformation and the ones in which the planes of succinimide and phenol are skewed to give the formation of an intramolecular H-bonding (Fig. 2c and Supplementary Table 25). On the contrary, in less polar solvent like chloroform where the H-bonding faces less intense competition from the bulk solvent molecules, the latter conformations were predicted to be slightly more stable in the cases of **1** and **2** (Supplementary Table 25).

**From high speed running to nearly a stop.** To explore the dynamic features of **1**–**3** in solution, we first carried out variable-temperature (VT) $^1$H NMR experiments in acetonitrile-$d_3$.

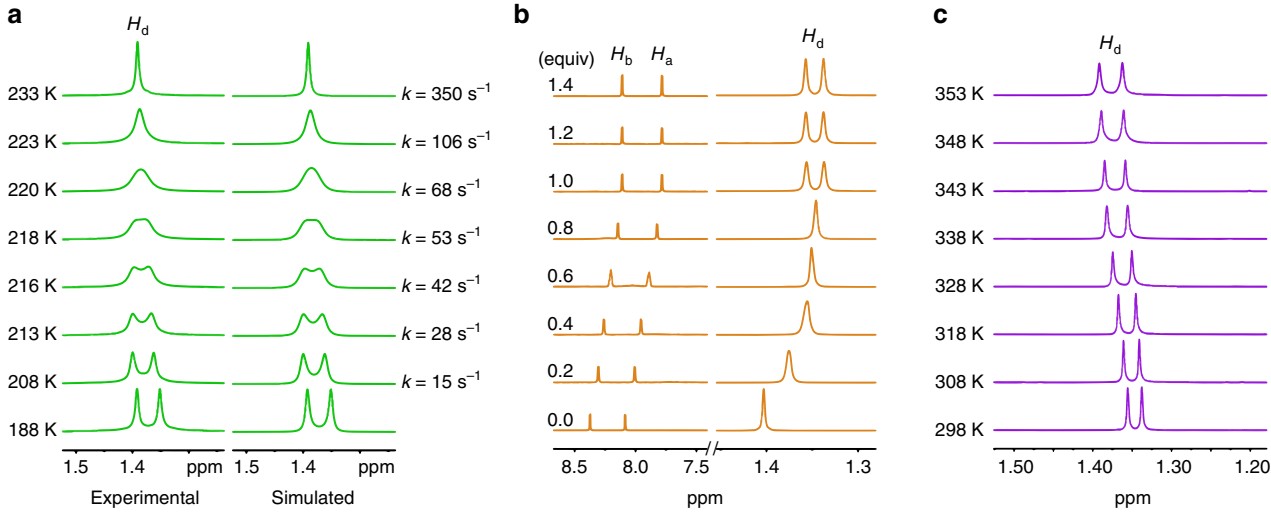

**Fig. 3** The native and the base-mediated rotation. **a** Experimental and simulated VT $^1$H NMR spectra (500 MHz) of **3** (4 mM) at the region of $H_d$ in acetone-$d_6$. The temperature (K) and the rate constants of the conformational isomerization ($k$, s$^{-1}$) are given for each trace. The rate constants were obtained by VT NMR line-shape analyses of the methyl resonances. **b** Part of $^1$H NMR spectra (500 MHz, 298 K) of **3** (4 mM) on titration of DBU in acetonitrile-$d_3$. The splitting of the $H_d$ signal indicates that rotation of **3** is highly-restricted after been fully deprotonated. **c** VT $^1$H NMR spectra (500 MHz, 298 K) of **3** (4 mM) deprotonated with excess (1.2 equiv) DBU in acetonitrile-$d_3$

**Table 1 Kinetic properties for the intramolecular rotation of 1–3 in acetonitrile-$d_3$ under different conditions**

| Studied system | Rotor | M$^+$ (equiv) | $\Delta H^{\ddagger}$ (kcal mol$^{-1}$) | $\Delta S^{\ddagger}$ (cal mol$^{-1}$ K$^{-1}$) | $\Delta G^{\ddagger}_{298\ K}$ (kcal mol$^{-1}$) | $k_{rot\ 298\ K}$ (s$^{-1}$)$^e$ | $\Delta G^{\ddagger}_{338\ K}$ (kcal mol$^{-1}$) | $k_{rot\ 338\ K}$ (s$^{-1}$)$^e$ |
|---|---|---|---|---|---|---|---|---|
| **1**$^a$ | **1** | / | 11.9 ± 0.9 | 2.0 ± 3.1 | 11.3 ± 0.1 | (1.6 ± 0.2) ×10$^4$ | 11.3 ± 0.2 | (1.7 ± 0.5) ×10$^5$ |
| **2**$^a$ | **2** | / | 11.7 ± 0.2 | 2.3 ± 0.7 | 11.0 ± 0.1 | (2.7 ± 0.2) ×10$^4$ | 10.9 ± 0.1 | (2.8 ± 0.3) ×10$^5$ |
| **3**$^a$ | **3** | / | 12.3 ± 0.1 | 6.7 ± 0.7 | 10.3 ± 0.1 | (8.5 ± 1.3) ×10$^4$ | 10.1 ± 0.1 | (1.1 ± 0.2) ×10$^6$ |
| **3-DBU**$^b$ | (**3**-H)$^-$ | / | 20.6 ± 0.7 | 3.2 ± 2.1 | 19.6 ± 0.1 | (1.3 ± 0.1) ×10$^{-2}$ | 19.5 ± 0.1 | 0.75 ± 0.01 |
| **3-DBU-LiClO$_4$**$^c$ | Li$^+$ ·(**3**-H)$^-$ | 2.0 | 15.1 ± 1.0 | −4.8 ± 3.1 | 16.6 ± 0.1 | 2.3 ± 0.7 | 16.8 ± 0.1 | 50 ± 3 |
| | | 1.0 | 14.0 | −9.0 | 16.7 | 1.7 | 17.1 | 28 |
| **3-tBuOLi**$^d$ | Li$^+$ ·(**3**-H)$^-$ | 1.0 | 15.6 ± 1.7 | −4.0 ± 5.1 | 16.8 ± 0.3 | 1.5 ± 0.6 | 17.0 ± 0.1 | 33 ± 7 |
| **3-DBU-NaClO$_4$**$^c$ | Na$^+$ ·(**3**-H)$^-$ | 2.0 | 14.6 ± 1.6 | −10.5 ± 4.5 | 17.7 ± 0.3 | 0.34 ± 0.04 | 18.1 ± 0.1 | 5.5 ± 0.3 |
| | | 1.0 | 15.6 | −8.5 | 18.1 | 0.16 | 18.4 | 3.7 |
| **3-tBuONa**$^d$ | Na$^+$ ·(**3**-H)$^-$ | 1.0 | 15.3 ± 1.4 | −9.4 ± 4.4 | 18.1 ± 0.1 | 0.17 ± 0.04 | 18.5 ± 0.1 | 3.6 ± 0.2 |
| **3-DBU-KClO$_4$**$^c$ | K$^+$ ·(**3**-H)$^-$ | 2.0 | 21.1 ± 2.4 | 5.1 ± 7.2 | 19.5 ± 0.3 | (1.6 ± 1.0) ×10$^{-2}$ | 19.4 ± 0.1 | 0.95 ± 0.06 |
| | | 1.0 | 21.8 | 7.1 | 19.6 | 1.2 × 10$^{-2}$ | 19.4 | 0.94 |
| **3-tBuOK**$^d$ | K$^+$ ·(**3**-H)$^-$ | 1.0 | 19.9 ± 1.3 | 1.3 ± 4.4 | 19.5 ± 0.1 | (1.6 ± 0.3) ×10$^{-2}$ | 19.4 ± 0.2 | 0.80 ± 0.29 |

The values were derived from Erying Plots obtained from 2D EXSY or VT $^1$H NMR experiments, carried out in acetonitrile-$d_3$ unless otherwise noted
$^a$In acetone-$d_6$
$^b$**3** in the presence of 1.2 equiv of DBU
$^c$**3** in the presence of 1.2 equiv of DBU and different amount of metal cations
$^d$**3** in the presence of 1.0 equiv of alkali metal *tert*-butoxides
$^e$Rotational speed at the given temperature, defined as half the rate of conformational isomerization of the rotor

However, the methyl signals of all of the studied molecules just slightly broadened upon cooling and showed no signs of decoalescence at all temperatures studied, even at the low-temperature limit of 233 K (Supplementary Method 4 and Supplementary Figs. 19–22). VT $^1$H NMR investigations (Supplementary Figs. 23–31) were then carried out in acetone-$d_6$, a solvent that has polarity just slightly smaller than that of acetonitrile-$d_3$ but a much lower melting point. In such cases, considerable broadening followed by splitting of the resonance of the methyl protons were clearly observed (Fig. 3a and Supplementary Figs 23, 26, and 29).

The rotational free energy of activations ($\Delta G^{\ddagger}$) of **1–3** in acetone-$d_6$ were calculated using the Eyring equation (Supplementary Fig. 25, 28, and 31)[52]. The $\Delta G^{\ddagger}$ value was 11.3 ± 0.1 kcal mol$^{-1}$ for **1** and 11.0 ± 0.1 and 10.3 ± 0.1 kcal mol$^{-1}$ for **2** and **3** at 298 K, respectively (Table 1). Overall, This suggests that the rotational barriers of the rotors are indeed influenced by the bond energy of the hydroxyl groups, although the influence in such a case is minimal. The DFT calculations (Supplementary Method 12), using the B3LYP/6-31+G(d,p) method and IEF-PCM-UFF solvent model, confirmed that the TS structures of the rotors adopt a planar geometry, stabilized by an intramolecular hydrogen-bonding between the hydroxyl O–H and the imide carbonyls (Fig. 2c and Supplementary Tables 24 and 29). The energy level of the TS, computed at the B3LYP/6-31G(d,p)/IEF-PCM-UFF+ZPVE levels of theory, is ca. 9, 7, and 8 kcal mol$^{-1}$ higher than that of the GS conformation for **1**, **2** and **3**, respectively, in three different solvents, including acetonitrile, acetone and chloroform (Supplementary Table 25). These gaps are solvent-independent and a little smaller (2–4 kcal mol$^{-1}$) than

the activation energies derived from the VT NMR experiments, which may be due to that the solvent model used in the calculations could not describe well the H-bonding between the solvent molecules and the rotors (Supplementary Note 1). Theoretically, H-bonding interactions between rotor and the solvent molecules would inevitably rival against the intramolecular one within **1–3** at the TSs, thus elevating the rotational barriers of the rotors. Nevertheless, the calculation results still indicate that the substituents on the phenol stator have impact on the movements of **1–3**, in agreement with the experimental results in acetone-$d_6$.

We next explored the properties of these rotor systems in the presence of 1,8-diazabicyclo[5.4.0]undec-7-ene (DBU) (Supplementary Method 5). DBU is known to be a strong, non-nucleophilic organic base (p$K$a = 12) with large steric hindrance[53]; and its conjugate acid, protonated DBU (DBU-H$^+$), is well-solvated in most of the common polar solvents. We hypothesized that the rotors would get completely deprotonated in the presence of DBU, and such deprotonation would generate a repulsion force between the resulting phenolate and the imide carbonyls, yielding restriction on the rotation for the rotors.

The deprotonations of the rotors in response to DBU were monitored by $^1$H NMR spectroscopy. As expected, considerable upfield shifts of all aryl protons were observed by treating **1–3** with one equivalent of DBU in acetonitrile-$d_3$ (Supplementary Figs 32–34), which is ascribed to the enhanced electronic density of phenolate oxyanions compared with the hydroxyl OH. In contrast to that in the cases of **1–3**, the substituents on phenol stator showed pronounced impact on the movements of the deprotonated molecules. Specifically, despite almost complete deprotonation in this situation (Supplementary Note 2), no obvious decoalescence of the methyl $^1$H NMR signals could be observed in the case of **1** and **2**. This indicates that even a very small amount of unprotonated rotors can induce an unhindered rotation of large amounts of deprotonated species, probably ascribed to the base-catalyzed proton exchange[54] between the phenolate oxyanions and the phenol O——H groups. Indeed, this is confirmed by the observation in the case of **3**, where the methyl protons clearly appeared as a doublet when 1 equiv of DBU was added (Fig. 3b). Despite that the methyl signal of **2** finally split until ca. 1.4 equiv of DBU was added, these results are still indicative of, on the one hand, the ability of DBU to slow down the rotation of the rotors and, on the other hand, the superiority of **3** to serve as a speed-tunable rotor over **1** and **2** ascribed to its lower p$K$a. Moreover, we found that, as a solvent, acetonitrile-$d_3$ possesses advantages over acetone-$d_6$. In the latter, the methyl signal of deprotonated **3** showed no apparent splitting (Supplementary Fig. 35). We thus mainly focused on **3** in the following studies, using acetonitrile-$d_3$ as the solvent.

Increasing temperature gave no obvious decoalescence of the methyl signal in the $^1$H NMR spectrum of the rotor in system of DBU-protonated **3** (denoted **3-DBU**) (Fig. 3c and Supplementary Fig. 36). The rotational free energy ($\Delta G^{\ddagger}$) of the rotor was thus measured by two-dimensional exchange NMR spectroscopy (2D EXSY) (Supplementary Figs. 37–42 and Supplementary Table 7)[55]. By extrapolating the Eyring plots in the 348–308 K range (Supplementary Fig. 43), the $\Delta G^{\ddagger}$ value is calculated to be ca. 19.6 ± 0.1 kcal mol$^{-1}$ at 298 K and 19.5 ± 0.1 kcal mol$^{-1}$ at 338 K (Table 1). The B3LYP/6-31+G(d,p)/IEF-PCM-UFF+ZPVE computations (Supplementary Method 12) for deprotonated **3** in acetonitrile in the absence of counter cations, (**3–H**)$^-$, predict the energy gap between the TS and the GS to be 22 kcal mol$^{-1}$ (Supplementary Tables 27–29), being consistent with the experimental value.

The rotational speed ($k_{rot}$) of **3-DBU** in acetonitrile-$d_3$, which is defined as the half rate of isomerization ($k$), was quantitatively assessed from the measured $\Delta G^{\ddagger}$ to be $(1.3 \pm 0.1) \times 10^{-2}$ s$^{-1}$ at 298 K (Table 1). On the other hand, using the measured rotational barrier obtained in acetone-$d_6$, we estimated the standard rate constant for the rotation of **3** in acetonitrile-$d_3$ at ambient temperature, $k_{rot} = (8.5 \pm 1.3) \times 10^4$ s$^{-1}$, which is 6–7 orders of magnitude higher than that in the case of **3-DBU**. Similar results were also obtained at elevated temperature such as 338 K (Table 1). These observations strongly indicate that a high-speed rotation of **3** could be highly restricted by treating it with base in a wide range of temperatures.

**Metal cations mediated mid speed rotations.** The electrostatic interaction between ligand (can be neutral or negatively charged) and a metal cation is one of important noncovalent interactions. In the field of supramolecular chemistry, such force has been widely utilized for cation recognition[56], molecular self-assembly[57], template-directed synthesis[58], and the construction of molecular machinery[48,59]. Since the findings presented above have demonstrated that H-bonding can stabilize the planer TS of **3**, we envisioned that the ligand-cation interaction would behave similarly on (**3–H**)$^-$. Besides, given larger cations have a lower charge to radius ratio and a larger steric hindrance compared to the smaller ones, we speculated that difference cation might possess a different capability of stabilizing the TS.

Three group 1 cations, including Li$^+$, Na$^+$, and K$^+$, with perchlorate as the counter anions, were chosen in our proof-of-concept studies (Supplementary Method 6, Supplementary Figs 44–123, and Supplementary Tables 8–19). In acetonitrile-$d_3$ at ambient temperature, adding two equiv of Li$^+$ cations to **3** which was previously deprotonated with excess (1.2 equiv) DBU (the obtained rotor was assigned as **Li$^+$·(3–H)$^-$** and the system denoted **3-DBU-LiClO$_4$**, similarly for that in the cases of other cations (Table 1)) provided no obvious changes in the $^1$H NMR spectrum. However, different from that of **3-DBU** (Fig. 3c), upon increasing the temperature, the methyl signals became coalescent and finally presented a sharp peak at 348 K (Supplementary Fig. 45). As expected, the corresponding $\Delta G^{\ddagger}$ at 298 K was calculated to be 16.6 ± 0.1 kcal mol$^{-1}$, which is ~3.0 kcal mol$^{-1}$ lower than that of **3-DBU** (Table 1). In the cases of **Na$^+$·(3–H)$^-$** and **K$^+$·(3–H)$^-$**, elevating the temperature did not give rise to a coalescence of the signals. The followed 2D EXSY NMR measurements suggested that the $\Delta G^{\ddagger}$ value at 298 K was 17.7 ± 0.3 and 19.5 ± 0.3 kcal mol$^{-1}$ for the rotor (**3–H**)$^-$ in the presence of Na$^+$ and K$^+$ cations, respectively (Table 1). The latter is almost identical to that of **3-DBU**, indicating K$^+$ has little influence on the rotation of (**3–H**)$^-$. Nevertheless, these data are still suggestive of a radius-dependent capability of the cations to decrease the rotational barrier of (**3–H**)$^-$. Similar results were also obtained in the cases at elevated temperatures (Table 1).

To confirm the cation-accelerating mechanism, we also measured the $\Delta G^{\ddagger}$ of **3** deprotonated with alkali metal *tert*-butoxides (Supplementary Method 6 and Supplementary Figs 56–68) (the obtained rotors were also assigned as **M$^+$·(3–H)$^-$** and the system denoted **3-*t*BuOM** (M = Li$^+$, Na$^+$, or K$^+$) (Table 1)). By such way, the influence of the counter-ions in previous system, DBU-H$^+$, on the rotation of the rotor could be completely ruled out. Given that the alkali metal *tert*-butoxides themselves are just sparingly (or slightly) soluble in acetonitrile-$d_3$, the samples under investigation were prepared by mixing **3** with equivalent lithium *tert*-butoxide or excess sodium or potassium *tert*-butoxides powder first, followed the addition of acetonitrile-$d_3$. The final homogeneous solutions of **3-*t*BuOM** in acetonitrile-$d_3$ were seen with visible precipitates of the excess

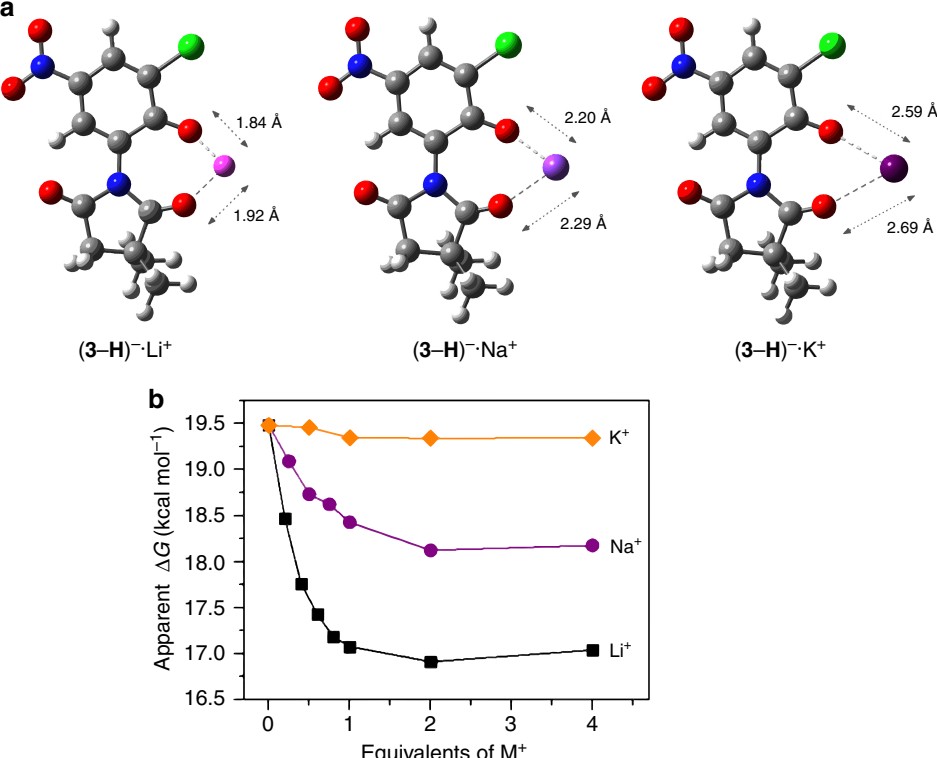

**Fig. 4** Rotation regulated by metal cations in acetonitrile-$d_3$. **a** DFT (B3LYP/6-31G(d,p)/IEF-PCM-UFF) optimized TS structures of the complexes of (**3**–**H**)$^-$ with Li$^+$, Na$^+$, and K$^+$ cations. The TSs adopt a planar geometry, stabilized by a O$^-$(phenolate)···M$^+$···O(carbonyl) coordination bridge. The lengths of the coordination bonds are given in each structure. Their values indicate that the coordination weakens with the increase of metallic radii. **b** Plots of the apparent free energy of activation ($\Delta G^{\ddagger}$) at 338 K for motion of (**3**–**H**)$^-$ versus the amount of Li$^+$, Na$^+$, and K$^+$ cations presented in the solution

alkali metal *tert*-butoxides at the bottom of the tube in the cases of Na$^+$ or K$^+$ cations. Integrating the $^1$H NMR signals of (**3**–**H**)$^-$ and the *tert*-butyl group (belonged basically to the produced *tert*-butanol) confirmed that there was only one equivalent of alkali metal cations presented in the rotor systems solution in all the cases. The $\Delta G^{\ddagger}$ value for the rotation of the complex rotors at 298 K in the case of **3**-*t***BuOLi**, **3**-*t***BuONa**, and **3**-*t***BuOK** was measured to be 16.8 ± 0.3, 18.1 ± 0.1, and 19.5 ± 0.1 kcal mol$^{-1}$ (Table 1), respectively, which is slightly larger than that of above corresponding DBU-H$^+$-presented system. For more reasonable comparison, we next measured the barriers for DBU-deprotonated **3** in the presence of only one equiv of Li$^+$, Na$^+$, and K$^+$ cations (Supplementary Figs 69–80), and the results coincided well with those in the systems of **3**-*t***BuOM** (Table 1). Such agreement was evident at 338 K as well, at which the differences in $\Delta G^{\ddagger}$ for a specified rotor in different systems could be also negligible (Table 1). All these demonstrated that the presence of DBU-H$^+$ in system of **3**-**DBU** has little influence on the dynamic of (**3**–**H**)$^-$, at least in the presence of the studied metal cations. It is noteworthy that, in the systems of **3**-**DBU**-**MClO$_4$** (M = Li$^+$, Na$^+$, or K$^+$), it is also possible that the added alkaline metals coordinate with DBU-H$^+$ to release the proton back to the phenoxide, thus decreasing the measured rotational barriers. However, this mechanism is not supported by the coincidence between above two contrast experiments. Furthermore, even if there is a trace amount (Supplementary Note 3) of proton released in this process, it will be inevitably neutralized by the excess amount of DBU presented in the system.

In addition to experimental findings, DFT calculations (Supplementary Method 12) also predicted very similar results. At B3LYP/6-31+G(d,p)-IEF-PCMUFF+ZPVE level of theory,

the energy level of TS of rotor-cation complex was predicted to be ca. 16, 18, and 19 kcal mol$^{-1}$ higher than that of the GS in the case of **Li$^+$** ·(**3**–**H**)$^-$, **Na$^+$** ·(**3**–**H**)$^-$, and **K$^+$** ·(**3**–**H**)$^-$, respectively (Supplementary Table 28). In the TS of the complexes, planar geometries stabilized by O$^-$(phenolate)···M$^+$···O(carbonyl) coordination bridges were observed (Fig. 4a and Supplementary Tables 27 and 29). Moreover, ascribed to the nature of the ligand-cation interactions, the metal cations in the TSs stay far from the oxygen atoms, and the length of the coordination bond increases with the ionic radius, confirming that the motion barrier of the complexes is controlled by the strength of the coordination but has nothing to do with the steric hindrance of the cations.

We further explored the influence of the amount of the metal cations presented on the movement of the deprotonated rotor. As shown in Fig. 4b and Supplementary Fig. 123, with the addition of Li$^+$ cations to system **3**-**DBU**, the apparent rotational barrier of the rotor at 338 or 298 K initially decreased dramatically and then lowered slowly (Supplementary Note 4). The asymptotic shape of the curve is suggestive of a catalysis effect of Li$^+$ with respect to accelerating the rotation of (**3**–**H**)$^-$ as well as the nature of a bimolecular interaction in the process. Similar phenomena have also been observed in other rotor systems[41,43,60,61]. It is noteworthy that the apparent barrier slightly increased on titration with Li$^+$ cations up to 2 equivalents[41], which was probably due to that the (**3**–**H**)$^-$ molecule tended to coordinate with two Li$^+$ cations simultaneously in the presence of a large amount of Li$^+$ cations. Titration of Na$^+$ cations gave very similar phenomena to that in the case of Li$^+$; while with K$^+$ cations, the apparent rotational barrier showed essentially no changes as expected.

By comparing the rotational speed of (**3**–**H**)$^-$ in the absence and the presence of different cations, we can obtain an intuitive

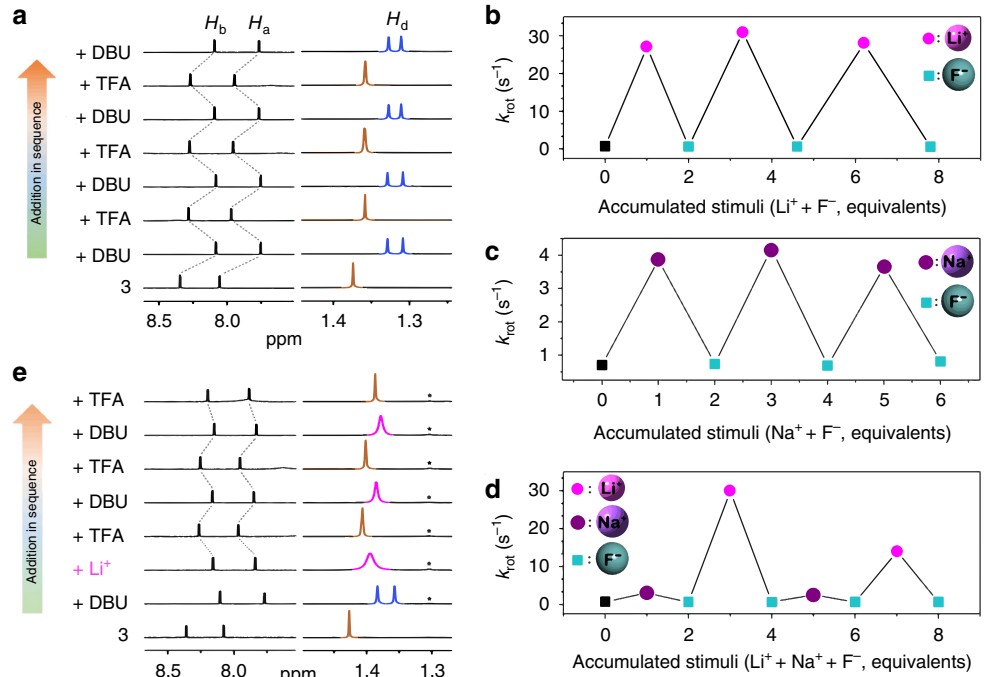

**Fig. 5** Inter-transformations of speed between different stages. **a** Changes in partial $^1$H NMR spectrum (500 MHz, 298 K, acetonitrile-$d_3$) of **3** (4 mM) upon alternate addition of DBU (1 equiv) and TFA (1 equiv). The highly reversible interconversion of the signal pattern of $H_d$ between a sharp coalescent peak and well resolved two signets demonstrates the reversibility in speed transformation between Stage-HS and Stage-Stop. **b–d** Changes in rotational speed of DBU-deprotonated **3** (4 mM) in acetonitrile-$d_3$ upon the cyclic addition of metal cations and fluoride anions (**a** addition of Li$^+$ and F$^-$ ions; **b** addition of Na$^+$ and F$^-$ ions; **c** addition of Na$^+$, F$^-$, Li$^+$, F$^-$, Na$^+$, F$^-$, Li$^+$, and F$^-$ ions in sequence). These changes gave an intuitive evidence of that reversible speed transformations among Stage-Li, Stage-Na, and Stage-Stop for the studied rotor could be realized by using ions stimuli. **e** Changes of $^1$H NMR spectra (500 MHz, 298 K, acetonitrile-$d_3$) of **3** (4 mM) upon the deprotonation with DBU (1 equiv), followed the addition of Li$^+$ cations (2 equiv) and alternate addition of TFA (1 equiv) and DBU (1 equiv). Tiny impurities (labelled with asterisk) presented in the DBU used in the experiment. Under the cyclic acid/ base stimuli, the $H_d$ signal of Li$^+$·(**3**–**H**)$^-$ alternately appeared as a sharp one and broad one, indicative of a reversible speed transformation between Stage-Li and Stage-HS. There is no need to remove the Li$^+$ cations in the system in steps of Stage-Li→Stage-HS transformtion becase that, once deprotonated **3** is neutralized, the interaction between the resultant **3** and Li$^+$ cations is negligible

proof on the accelerating capability of the cations. As can be derived from the data shown in Table 1, at 298 K, there is a nearly 177-fold and 26-fold acceleration of the rotation speed due to the presence of Li$^+$ and Na$^+$, respectively; while at 338 K, a 66-fold and 7-fold acceleration in the case of Li$^+$ and Na$^+$, respectively, is observed. Therefore, along with the high-speed rotation of **3** itself in solution, a molecular rotor system with the rotational speed that could be turned to at least four different stages, including a high-speed stage (denoted Stage-HS), a highly-restricted rotation stage (denoted Stage-stop), and two mid-speed stages (denoted Stage-Li and Stage-Na, respectively), is thus obtained (Fig. 1b).

**Speed transformations between different stages**. We next explored the possibility of transforming the rotational speed of the studied rotor between different stages (Supplementary Methods 7–11, Supplementary Figs 124–148, and Supplementary Tables 20–23). Such property should be one of the most essential characteristics for a practical molecular rotor.

By using acid/base mediation, transformation between Stage-HS and Stage-Stop could be achieved in a highly reversible manner (Supplementary Method 7). Except that deprotonation of **3** in acetonitrile-$d_3$ with DBU gave a restriction of the rotation for the rotor as described above, neutralization of the **3-DBU** solution with trifluoroacetic acid (TFA, p$K_a$=0.52) reproduced a sharp coalescent resonance of the methyl protons, indicating that

the rotor restarted to rotate fast (Fig. 5a). After several cycles of alternate addition of DBU and TFA, the dynamic of rotor **3** still showed very good regulability.

Both LiF and NaF are hardly soluble in acetonitrile due to their high lattice energies (The solubility of LiF and NaF is reported to be 0.09 ± 0.01 and 0.029 ± 0.007 mM, respectively, in acetonitrile at 24–25 °C)[62]. In an effort to demonstrate the reversibility of cations-stimuli-responses, fluoride anions were first added into the system of **3-DBU-LiClO$_4$** to precipitate (defunctionalize) Li$^+$ cations and the dynamic of the rotor was monitored (Supplementary Method 8). At 338 K, upon the addition of 1.0 equiv of tetraethylammonium fluoride (TEAF) to the solution of **3** in the presence of excess (2.0 equiv) DBU and 1.0 equiv of LiClO$_4$ (Supplementary Note 5), a precipitate appeared. Meanwhile, a clear decoalescence of the methyl signal in the $^1$H NMR spectrum was observed (Supplementary Fig. 125), indicating that the Li$^+$ cations became ineffective after been precipitated. Further addition of 1.0 equiv of Li$^+$ cations gave recovery of the singlet pattern of the signals. After repeating the cycle of Li$^+$→F$^-$ stimuli for three times, there was no apparent deterioration in the spectral pattern of the methyl protons for the base-regulated rotor; however, a slightly splitting of the methyl signal of the regenerated **Li$^+$·(3–H)$^-$** was observed. The calculated rotational speeds confirmed that the rotation of **Li$^+$·(3–H)$^-$** slowed down apparently with the increase of number of stimuli cycles, probably ascribed to the partial hydrolysis of Li$^+$ cations during the circling experiment. As expected, with the addition of extra more Li$^+$ and

F$^-$ ions in each stimuli cycle, a reversible inter-transformation between Stage-Stop and Stage-Li was achieved (Fig. 5b and Supplementary Fig. 130). The highly reversible cations-stimuli-responses was also observed in the case of **Na**$^+$·**(3–H)**$^-$ (Supplementary Method 9), in which alternate addition of one equivalent of F$^-$ and Na$^+$ ions gave the methyl signal pattern switched reversibly between a sharp doublet and an obtuse one (Supplementary Fig. 134). In contrast to that in the case of Li$^+$ cations, no apparent hydrolysis of Na$^+$ cations was observed, which may due to the lower pK$_b$ and lattice energy of NaOH compared to LiOH. Reversibility of the inter-transformation between Stage-Stop and Stage-Na was also confirmed by the calculated rotational speeds that quantitatively assessed by the EXSY experiments (Fig. 5c).

Fluoride anions can also be utilized to tune the rotational speed switching among Stage-Li, Stage-Stop, and Stage-Na. For rotor system of **3-DBU-NaClO₄** (prepared by mixing of **3** with excess DBU and 1.0 equiv of NaClO₄) in acetonitrile-$d_3$, after the Na$^+$ cations in the system were defunctionalized with F$^-$ anions (to give a Stage-Stop), addition of 1.0 equiv of Li$^+$ cations gave the system showing the same dynamic feature as that of the original Li$^+$-stimulated one (Stage-Li) (Fig. 5d, Supplementary Method 10, and Supplementary Fig. 141). Again, these Li$^+$ cations can be removed from the solution with F$^-$ anions (to Stage-Stop), and the followed addition of 1.0 equiv of Na$^+$ cations gave the rotor re-rotating with speed at Stage-Na (Fig. 5d). Upon a two-cycle repeat with the above compounds, the stages Stage-Na and Stage-Stop kept showing good recoverability, whereas speed at Stage-Li decreased a little compared to that at the normal state for **Li**$^+$·**(3–H)**$^-$ (Fig. 5d), which again might be due to the partial hydrolysis of Li$^+$ cations and can theoretically be amended by adding more Li$^+$ cations.

Given that there is negligible interaction between **3** and alkali metal cations, switching between a high-speed rotation and mid-speed ones for the studied systems can be realized readily by using acid/base stimuli. As an example, neutralizing deprotonated **3** in the presence of 2.0 equiv of Li$^+$ cations with TFA gave a sharp $^1$H NMR resonance of the methyl protons of the rotor (indicative of a fast rotation of **3**); and the followed basification with DBU rebuilt a rotor system of **Li**$^+$·**(3–H)**$^-$, resulting in a board signet of the methyl signal (Fig. 5e and Supplementary Method 11). High reversibility of the inter-transformation can also be observed in the cyclic stimuli experiments.

## Discussion

In summary, we have constructed a molecular rotor, the rotational speed of which in the slow-to-fast range could be reversibly modulated to at least four stages by using acid/base and metal cations as the triggers. The speed tunability is achieved by tuning the stability of the transition state of the rotor to varying degrees using H-bonding, electrostatic repulsion and different coordination interactions. The key point in the speed regulation is the combined utilization of several different supramolecular interactions, in particular different metal coordinations, to accomplish the goal. Overall, the concept utilized in our research offers a new avenue to constructing multi-speed molecular rotors. Besides, given the diversity of metal-coordination interactions, there is no doubt such concept could be used, in the future, to further improve the properties of such multistage rotational-speed-changing rotor or similar systems, by applying stronger metal–ligand coordination interactions, thus promoting the application of molecular rotors in constructing nanomachines with elaborate functions.

## Methods

**$^1$H NMR line shape analyses**. Throughout our kinetics investigations, rotational barriers of rotors lower than 18 kcal mol$^{-1}$ were measured via variable-temperature (VT) $^1$H NMR followed by line-shape analyses (LSA)[63]. Calculations of the exchange rates between a pair of diastereotopic methyl protons were performed by LSA of the experimental methyl $^1$H NMR signals using DNMR procedure implemented in Topspin 2.1 software (Bruker BioSpin Group) package. Despite that the $^1$H NMR spectra of some samples might be recorded in a wide range of temperature, to meet the requirement of total line-shape analysis, only spectra recorded in intermediate exchange region were used for simulation. In the fittings, the temperature dependent $^1$H NMR signals were simulated within a one-spin system undergoing a two-site exchange model. The calculated exchange rates (rates of conformational isomerization) ($k$, s$^{-1}$) are given for each trace on the corresponding NMR spectra in the main text and the Supplementary Information.

**2D EXSY Investigation**. For rotational barriers those are higher than 18 kcal mol$^{-1}$, the exchange rates ($k$, s$^{-1}$) of the dynamic systems were measured via 2D EXSY, based on the cross peak to diagonal peak intensity ratio at a single 2D EXSY NMR spectrum. The exchange rate ($k$, s$^{-1}$) between two non-coupled spins of the diastereotopic methyl groups (denoted spin A and B, respectively) were estimated using the following equation:

$$k = \frac{1}{t_m} \ln\frac{r+1}{r-1}, \tag{1}$$

where

$$r = \frac{I_{AA} + I_{BB}}{I_{AB} + I_{BA}}, \tag{2}$$

and $t_m$ is the mixing time.

In the investigations, the investigation temperature intervals were carefully selected to make sure that the chemical exchange is observable on the NMR timescale.

**Kinetic parameters**. Using the exchange rates ($k$, s$^{-1}$) obtained from LSA of the VT $^1$H NMR spectra and 2D EXSY spectra, the enthalpic ($\Delta H^\ddagger$) and entropic ($\Delta S^\ddagger$) contributions to the transition state were calculated from Eyring plots (see equation below).

$$\ln\frac{k}{T} = -\frac{\Delta H^\ddagger}{R}, \frac{1}{T} + \ln\frac{k_B}{h} + \frac{\Delta S^\ddagger}{R}, \tag{3}$$

where $k$ is the exchange rate constant, $T$ the absolute temperature, $\Delta H^\ddagger$ the enthalpy of activation, $R$ the universal gas constant, $k_B$ the Boltzmann constant, $h$ the Planck's constant and $\Delta S^\ddagger$ the entropy of activation.

Fortunately, in all cases, the Eyring plots displayed a high degree of linearity in the range of temperature examined. The obtained $\Delta H^\ddagger$ and $\Delta S^\ddagger$ are summarized in Table 1 and Supplementary Tables 5–6 (Supplementary Note 6).

Basically, the free energy of activation ($\Delta G^\ddagger$) was determined from,

$$\Delta G^\ddagger = \Delta H^\ddagger - T\Delta S^\ddagger \tag{4}$$

using the activation enthalpy ($\Delta H^\ddagger$) and the entropy ($\Delta S^\ddagger$) derived from the Eyring plots. The $\Delta G^\ddagger$ values at two different temperatures, including 298 and 338 K, for all the studied rotor systems were calculated. A linear extrapolation method was used for the calculations, based on an assumption that the enthalpy and the entropy terms remain constant over a wide temperature range. Such strategy has been widely used in the field of molecular kinetics studies[16,43,64,65].

To reduce the systematic errors, the overall rates of chemical exchanges ($k$, s$^{-1}$) were obtained by Eyring–Polanyi equation,

$$k = \frac{K_B T}{h} e - \frac{\Delta G^\ddagger}{RT} \tag{5}$$

rather than directly read the ones derived from LAS and 2D EXSY at a given temperature.

Since the diastereotopic methyl protons undergo chemical exchanges twice in every rotational cycle for the rotor systems, the rotational speed ($k_{rot}$, s$^{-1}$) of the systems is defined as half the rate of chemical exchange ($k$, s$^{-1}$) between the pair of diastereotopic methyl protons of the molecules, i.e., $k_{rot} = 0.5 k$. The obtained values for the rotational speeds are summarized in Table 1.

**Data availability**. The authors declare that the all data supporting the findings of this study are available within this article and Supplementary Information files, and also are available from the authors upon reasonable request. CCDC 1448538,

1448541, and 1448542 contain the crystallographic data for this paper. These data can be obtained free of charge from The Cambridge Crystallographic Data Centre via www.ccdc.cam.ac.uk/getstructures.

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

## Acknowledgements

We acknowledge the financial supports from the National Natural Science Foundation of China (21332008, 21572023 and 21672026) and the 973 Program (2015CB856502).

## Author contributions

Y. Wang and H.J. conceived of and designed the experimental studies and supervised the work. Y. Wu, G.W. and Q.L. conducted rotors synthesis and performed dynamic characterization. J.X. provided guidance for EXSY studies. Y. Wang and Y. Wu designed and conducted computational studies. Y. Wang wrote the paper.

## Additional information

**Competing interests:** The authors declare no competing interests.

