## [Peer Review File · Nature Communications]

Reviewers' comments:

Reviewer #1 (Remarks to the Author):

Clearly the first round of reviews has drastically improved the quality of this paper. Having said that, and in agreement with one of the other reviewers, this research still lacks novelty and is an incremental advancement over the Shimidzu other works mentioned in the 1st round of review. After reading the response and the new version I still think that this paper is suited for ChemCommun (or a similar journal).

Reviewer #2 (Remarks to the Author):

This manuscript reports a new molecular rotor of four stages of rotation rate operated by chemical stimuli acids/bases and alkali metal ions/fluoride ions. The strategy of using metal coordination to stabilize the rotational transition state for such a control of the rotation rate is novel. In addition, This revised manuscript carefully tackled the problems mentioned by all the reviewers and thus the quality of the current form is much improved. Therefore, its publication in Nature Commun is recommended after considering the following issue:

In line 256, the authors stated, "integration of ^1H NMR signals confirmed that there was only one equivalent of alkali metal cations presented in the rotor systems solution in all the cases, regardless of whether 1.3 or 2.0 eq of sodium or potassium tert-butoxide was added initially." How is the concentration of the metal cation determined by the integration of ^1H NMR signals? What particular signal is responsible for this conclusion? This requires further elaboration.

Reviewer #3 (Remarks to the Author):

The authors have taken considerable effort to address all of the questions and suggestions of the three reviewers. In particular, the authors have remeasured most of the rotational barriers, which has greatly improved the accuracy of the data and the certainty of the conclusions.

The authors have also addressed the primary concern of this reviewer, which was to clarify the novel and unique aspects of this project. Specifically, this reviewer was concerned that the objective of designing and studying a multi-speed molecular rotor had previously been reported. There were several examples in the literature of molecular rotors that could be controlled by adding increasing concentrations of the guest-stimuli. However, the authors make excellent arguments that previous rotors have been primarily two speed rotors (fast and slow). Thus, increasing addition of guest-stimuli to these systems would give the appearance of stepwise increases of the rate of rotation. But, this appearance was due to the average speed changing due to changes in the ratio of slow and fast rotors. The authors provide solid evidence that their system is truly a multi-speed motor as the proton and different metal ions accelerate the rotor to varying degrees.

Therefore, this manuscript meets the novelty and importance requirements for publication in Nature Chemistry. However, there are a number of issues that need to be addressed before publication. These are listed below in order of importance

1. The writing and English in this manuscript makes it very difficult to access and understand. A major problem is the use of non-idiomatic terms or phrases. Some examples include: ultraminiturized, speed regulation, ultraslow, polarized stator, midspeed, less suffering the competition, in such case, even there is, fast-speed, re-constructed. There are many more. Also, many of the new passages that have been inserted into this revision (highlighted in yellow) have grammatical errors and run on sentences.
2. Further exploration of the literature of molecular rotors identified two additional references that need to be included and highlighted in the text. The first is early work by Rebek which is summarized in a review paper (Acc. Chem. Res. 1984,17, 258-264.) This work is directly relevant and overlapping with this work as Rebek describes the modulation of the rotational barrier of bipyridines by the addition of metal ion and proton guests, which bind to and stabilize the planar transition state. The second reference is a recent publication by Shimizu in which the rotation speed of the same molecular rotor framework (N-arylimide) is modulated by hydrogen bonding (Chem. Commun., 2017, 53, 12469-12472)
3. The authors need to quantitatively define fast, ultrafast, slow, mid-speed and ultraslow rotation speeds. Given the large number of different speeds, the authors may want to have a more systematic method of naming these speeds.
4. The authors do not need to repeatedly show the VT ¹H NMR data (Figure 3 and 4). This can be placed in the SI. A much more helpful and convincing figure would be an expanded version of the plot shown in Figure 4c in which the titration curves for Li, Na, K, and proton are shown on the same plot.

5. In figure 2, the authors show the wrong isomer of the TS and hydrogen bonded intermediate. The lower TS and intermediate would have phenol hydrogen bonded to the imide carbonyl that is connected to a CH₂ group and not to the more bulky C(CH₃)₂ group.

6. In figure 5d, can the authors explain why the second addition of Li⁺ does not yield the same rotational speed and the first addition of Li⁺?

Response to reviews:

Reviewer #1: We thank the reviewer for the comments. Responses are provided below.

Reviewer #1 (Remarks to the Author):

Clearly the first round of reviews has drastically improved the quality of this paper. Having said that, and in agreement with one of the other reviewers, this research still lacks novelty and is an incremental advancement over the Shimidzu other works mentioned in the 1st round of review. After reading the response and the new version I still think that this paper is suited for ChemCommun (or a similar journal).

Answer: We appreciate the reviewer for the positive comment on our work in improving the quality of our manuscript with the help of all the three reviewers, but want to re-emphasize on the novelty of our research.

There is no doubt that one of key issues in the construction of truly-functioning molecular machines is to control the motion of the machine molecules precisely and effectively. As regard to molecular rotors, precise control means at least the rotational speed could be continuously modulated in a wide range of frequencies, which enables the rotors to perform tasks like those of their macroscopic counterpart in ordinary machines. However, to establish such molecular rotors is still formidably challenging: to date, most of the reported molecular rotors are "ON-OFF" systems; three isolated cases involve finely-tuning the rotation, but the speed could only be modulated in a narrow range of frequencies. A molecular rotor with the rotational speed in the slow-to-ultrafast range could be finely tuned to several different stages, as that we can often find in macroscopic rotors, remains elusive.

In our manuscript, we reported such a stimuli-responsive molecular rotor, the rotational speed of which in the slow-to-fast range could be modulated to at least four stages. The speed tunability is achieved by tuning the stability of the transition state of the rotor to varying degrees using H-bonding, electrostatic repulsion and coordination interactions with acid/base and metal cations as the triggers.

Overall, the most notable merits of our work lie in:

1) Performance: *The studied multistage rotational-speed-changing rotor represent the first example of molecular rotors showing multi-stage rotational rate switching and tuning in 6-7 orders of magnitude from a fast to a slow (stop) state at a given temperature.* Such range of changing in frequency, to a high degree, resembles those of the motion of macroscopic rotors in ordinary machine, for examples, the rotational systems in cars and electronic fans. In this regard, Shimidzu's rotor rotates faster

upon protonation, which is fundamentally just a "ON-OFF" system.

2) Methodology: For the first time, the rotational barrier (and hence the rotational frequency) of a real molecular rotor was finely tuned to several discontinuous stages by lowering the energy level of the transition state through the formation of intramolecular interactions (including H-bonding, and different coordination interactions) between the stator and the rotator, with varying interaction strengths, to stabilize the rotational transition state in varying degrees. The key point in the concept is the combined utilization of several different supramolecular interactions, in particular several different metal coordination, to accomplish the goal. Shimidzu's research used acid to form intramolecular H-bonding to stabilize the transition state. This strategy, i.e., using just one kind of supramolecular interactions, certainly could not be used for constructing a multistage rotational-speed-changing rotor, even if the strategy is expanded to that "tuning the strength of the intramolecular H-bonding, by using various H-bonding guests to provide different H-bonding competition" (The effectiveness of the expanded strategy is suspicious because H-bonding competitive guests may just change the content but not the rotational speed of the rotor species presenting in the system, although the apparent (measured) rotational barrier changes upon the addition of the guests). By the same token, utilizing only one kind of metal coordination could not fulfil the task as well, even if that various coordination competitive guests are used. Given the diversity of metal-coordination interactions, the concept utilized in our research offers a new avenue to constructing multi-speed molecular rotors.

Our research is essentially not an incremental advancement over the Shimidzu's work; the underlying concepts are rationally designed with the goals and the drawbacks of other peoples' works in mind. There is no doubt our concepts could be used, in the future, to further improve the properties of such multistage rotational-speed-changing rotor or similar systems, by applying stronger metal-ligand coordination interactions.

Reviewer #2: We thank the reviewer for the supportive comments and insightful comments. Responses are provided below.

Reviewer #2 (Remarks to the Author):

This manuscript reports a new molecular rotor of four stages of rotation rate operated by

chemical stimuli acids/bases and alkali metal ions/fluoride ions. The strategy of using metal coordination to stabilize the rotational transition state for such a control of the rotation rate is novel. In addition, This revised manuscript carefully tackled the problems mentioned by all the reviewers and thus the quality of the current form is much improved. Therefore, its publication in Nature Commun is recommended after considering the following issue:

In line 256, the authors stated, " integration of ^1H NMR signals confirmed that there was only one equivalent of alkali metal cations presented in the rotor systems solution in all the cases, regardless of whether 1.3 or 2.0 eq of sodium or potassium tert-butoxide was added initially." How is the concentration of the metal cation determined by the integration of ^1H NMR signals? What particular signal is responsible for this conclusion? This requires further elaboration.

Answer: We are very grateful to the reviewer for his/her affirmative comments on our work in improving the quality of our manuscript under the help of the reviewers. We also thank the reviewer for suggesting the clarification.

Concentrations of the metal cations were determined based on the ratio of the integral of the *t*-butyl signal located around 1.20 ppm on the ^1H NMR spectra to the signals of deprotonated **3**. The *t*-butyl signal was basically attributed to *t*-butanol, a product of the reaction of **3** with alkali metal *t*-butoxide.

We added a brief explanation in the revised manuscript (page 11) to address this issue: "Integration of the ^1H NMR signals of (3-H)⁻ and the tert-butyl group (belonged basically to the produced tert-butanol) confirmed that there was only one equivalent of alkali metal cations presented in the rotor systems solution in all the cases,

Reviewer #3: We thank the reviewer for the supportive comments and insightful suggestions. Responses are provided below.

Reviewer #3 (Remarks to the Author):

The authors have taken considerable effort to address all of the questions and suggestions of the three reviewers. In particular, the authors have remeasured most of the rotational barriers,

which has greatly improved the accuracy of the data and the certainty of the conclusions.

The authors have also addressed the primary concern of this reviewer, which was to clarify the novel and unique aspects of this project. Specifically, this reviewer was concerned that the objective of designing and studying a multi-speed molecular rotor had previously been reported. There were several examples in the literature of molecular rotors that could be controlled by adding increasing concentrations of the guest-stimuli. However, the authors make excellent arguments that previous rotors have been primarily two speed rotors (fast and slow). Thus, increasing addition of guest-stimuli to these systems would give the appearance of stepwise increases of the rate of rotation. But, this appearance was due to the average speed changing due to changes in the ratio of slow and fast rotors. The authors provide solid evidence that their system is truly a multi-speed motor as the proton and different metal ions accelerate the rotor to varying degrees.

Therefore, this manuscript meets the novelty and importance requirements for publication in Nature Chemistry. However, there are a number of issues that need to be addressed before publication. These are listed below in order of importance

1. The writing and English in this manuscript makes it very difficult to access and understand. A major problem is the use of non-idiomatic terms or phrases. Some examples include: ultraminiturized, speed regulation, ultraslow, polarized stator, midspeed, less suffering the competition, in such case, even there is, fast-speed, re-constructed. There are many more. Also, many of the new passages that have been inserted into this revision (highlighted in yellow) have grammatical errors and run on sentences.

Answer: We are very grateful to the reviewer for his/her affirmative comments on our work in improving the quality of our manuscript under the help of the reviewers; we also thank this reviewer for pointing out those inappropriate expressions.

As our additional effort, we have carefully retouched the whole manuscript with respect to the language and the grammar. Hopefully the English has been greatly improved for publication. In the revised manuscript, most of the non-idiomatic phrases have been changed, replaced or deleted; while some of them are retained, considering that they have been also used by the others in this field in their papers

and that it is really difficult for us to find better words or phrases to replace them. Some changes are listed in the following table in detail.

Previous Text	Action	New Text	Examples of Literatures or Notes
ultraminaturized machines	changed	nanomachines	
speed regulation	remained		Nat. Chem. 9, 480–486 (2017); Energy Conv. Manag. 52, 1252–1257 (2011); Patent US 3,512,067.
ultraslow	changed	slow	
a polarized phenol stator	changed	a stator of phenol with the O–H group polarized	
mid-speed; mid-speed range	remained		J. Turbomach. 116(2), 194-201 (1994) US Patent 9,629,584,
where H-bonding is less suffering the competition from the solvent molecules	changed	where the H-bonding faces less intense competition from the bulk solvent molecules	
in such case	changed	in such a case	
even there is	changed	even if there is	
fast-speed	changed	high-speed	
re-constructed	changed	rebuilt	
built up	changed	construct	
shape persistent	changed	shape-persistent	

2. Further exploration of the literature of molecular rotors identified two additional references that need to be included and highlighted in the text. The first is early work by Rebek which is summarized in a review paper (*Acc. Chem. Res.* 1984,17, 258-264.) This work is directly relevant and overlapping with this work as Rebek describes the modulation of the rotational barrier of bipyridines by the addition of metal ion and proton guests, which bind to and stabilize the planar transition state. The second reference is a recent publication by Shimizu in which the rotation speed of the same molecular rotor framework (*N*-arylimide) is modulated by hydrogen bonding (*Chem. Commun.*, 2017, 53, 12469-12472)

Answer: We would like to thank the reviewer for these two additional highly relevant references. We have added these two important literatures into the revised manuscript (reference 22 and 49). Despite of aiming at a different goal, Rebek's research utilized metal cation and proton guests to stabilize the TS for modulating the racemization rate of the molecule, which is highly relevant with our research. The

Shimizu's new research focused again on the molecular rotor, using different guests with different H-bonding abilities to tune the rotational speed of the molecule, as we envisaged in our previous response letter. Constructive and meaningful experiments are involved in the research; but we think more supporting evidences are needed to rule out the alternative mechanism in which different H-bonding guests just change the content but not the rotational speed of the rotors at the native state and those with guest H-bonded.

Therefore, in addition to citing the two important literatures into the revised manuscript, we also added the following comments to highlight them: "*Rebek has reported that metal complexation and protonation gave stabilizing the TS of a bridged bipyridyl derivative thus being able to modulate the racemization rate of the bipyridyl.*" (on Page 4); and that "*More recently, Shimizu verified that apparent rotational barrier of a rotor could be tuned by using guests with different H-bonding abilities to disrupt the intramolecular H-bond that catalyzes the rotation of the rotor, but corroboration may be still required to identify whether addition of the guests changed the rotational speed of the rotor or just the ratio of the rotor molecules at the native state to those with the quest H-bonded.*" (on Page 3)

3. The authors need to quantitatively define fast, ultrafast, slow, mid-speed and ultraslow rotation speeds. Given the large number of different speeds, the authors may want to have a more systematic method of naming these speeds.

Answer: We thank the reviewer for this insightful/helpful suggestion.

In the revised manuscript, we deleted the expression and the speed class of "ultraslow", considering that: 1) "ultraslow" is a non-idiomatic term as pointed out by the reviewer; 2) in the view of machinery, there is typically not that much difference for a rotor running in frequency of 10^{-4} Hz and of 10^{-2} Hz (notably, for rotors in macroscopic machines, running in such frequency means "almost stop"), so that using just term "slow" might be more reasonable.

We tentatively define ultrafast, fast, mid-speed and slow rotation speeds as follows:

ultrafast	$> 10^7$ Hz
fast	$10^3 - 10^7$ Hz
mid-speed	$10^{-1} - 10^3$ Hz
slow	$< 10^{-1}$ Hz

To address this issue in our revised manuscript, we added the comments as follows: “Ideally, the molecular rotors are similarly of multistage rotational rates that overall cover a wide range of frequency to enable them to operate at slow ($< 10^{-1}$ Hz), intermediate ($10^{-1} - 10^3$ Hz), fast ($10^3 - 10^7$ Hz) or ultrafast ($> 10^7$ Hz) speeds as required.” (Page 1)

4. The authors do not need to repeatedly show the VT 1H NMR data (Figure 3 and 4). This can be placed in the SI. A much more helpful and convincing figure would be an expanded version of the plot shown in Figure 4c in which the titration curves for Li, Na, K, and proton are shown on the same plot.

Answer: Figure 4 was changed accordingly to address the reviewer’s concerns. As suggested, we have transferred the experimental and simulated VT 1H NMR spectra of **3-DBU-LiClO₄** (Figure 4a in the previous manuscript) to the Supplementary Information (supplementary fig. 45). Furthermore, as suggested, we have supplemented the 2D EXSY titration experiments for Na⁺ and K⁺ and added the titration curves to the figure of 4c (now figure 4b in the revised manuscript).

Experiment on the titration of proton has not been taken, considering that it is a reverse process of the experiment of DBU-titration of **3** (Supplementary Fig. 34). More importantly, part of such experiment is hampered by the comparatively high freezing point of acetonitrile-*d*₃ (M.p., -46 °C) to carry out. In our research, we used the rotational barrier in acetone-*d*₆ to estimate that in acetonitrile-*d*₃ for **3**. Figure 3 shows VT ¹H NMR of **3**, ¹H NMR DBU-titration of **3** and VT NMR of deprotonated **3**. We put these spectra there because we think, by this way, the readers can get some intuitive impressions about that to what extent the deprotonation changes the dynamic

of **3** (given that the difference in VT ¹H NMR spectra between **3** and deprotonated **3** is really huge) and that how effective by this means. We prefer to keep the current style but can change if the reviewers/editors still request us to.

5. In figure 2, the authors show the wrong isomer of the TS and hydrogen bonded intermediate. The lower TS and intermediate would have phenol hydrogen bonded to the imide carbonyl that is connected to a CH₂ group and not to the more bulky C(CH₃)₂ group.

Answer: We thank the reviewers for pointing out the error. This has been corrected in the revised manuscript.

6. In figure 5d, can the authors explain why the second addition of Li⁺ does not yield the same rotational speed and the first addition of Li⁺?

Answer: We attribute this to the partial hydrolysis of Li⁺ cations. In our manuscript, we commented on this phenomenon as follows: “Upon a two-cycle repeat with the above compounds, the stages Stage-Na and Stage-Stop kept showing good recoverability, whereas speed at Stage-Li decreased a little compared to that at the normal state for Li⁺·(3-H)⁻ (Fig. 5d), which again might be due to the partial hydrolysis of Li⁺ cations and can theoretically be amended by adding more Li⁺ cations.”

REVIEWERS' COMMENTS:

Reviewer #2 (Remarks to the Author):

This revised manuscript has been further improved by considering all the reviewers' comments. The authors answered all my questions. I do not have further comment on this manuscript. It can be accepted now.

Reviewer #3 (Remarks to the Author):

The authors have addressed all of the concerns raised by this reviewer. The manuscript content and presentation has improved considerably with each revision. Therefore, this manuscript is recommended for publication in Nature Communications.

Response to reviews:

Reviewer #2 (Remarks to the Author):

This revised manuscript has been further improved by considering all the reviewers' comments. The authors answered all my questions. I do not have further comment on this manuscript. It can be accepted now.

Reviewer #3 (Remarks to the Author):

The authors have addressed all of the concerns raised by this reviewer. The manuscript content and presentation has improved considerably with each revision. Therefore, this manuscript is recommended for publication in Nature Communications.

Answer: We are very grateful to the reviewers for these positive comments.